# Effect of One Step Solid State Reaction Route on the Semiconductor Behavior of the Spinel (NI, Co, and Mn)O_4_ to Be Used as Temperature Sensor

**DOI:** 10.3390/s23125380

**Published:** 2023-06-06

**Authors:** Daehyeon Ko, Sungwook Mhin

**Affiliations:** Department of Advanced Materials Engineering, Kyonggi University, Suwon 16227, Republic of Korea; godh3134@naver.com

**Keywords:** electroceramics, negative temperature coefficient, spinel, temperature sensor

## Abstract

Achieving carbon neutrality is important to solve environmental problems and thus requires decarbonizing manufacturing processes to reduce greenhouse gas emissions. The firing of ceramics, including calcination and sintering, is a typical fossil fuels-driven manufacturing process that requires large power consumption. Although the firing process in manufacturing ceramics cannot be eliminated, an effective firing strategy to reduce processing steps can be a choice to lower power consumption. Herein, we suggest a one-step solid solution reaction (SSR) route to manufacture (Ni, Co, and Mn)O_4_ (NMC) electroceramics for their application in temperature sensors with negative temperature coefficient (NTC). Additionally, the effect of the one-step SSR route on the electrical properties of the NMC is investigated. Similar to the NMC prepared using the two-step SSR route, spinel structures with dense microstructure are observed in the NMC prepared via the one-step SSR route. Based on the experimental results, the one-step SSR route can be considered as one of the effective processing techniques with less power consumption to manufacture electroceramics.

## 1. Introduction

The semiconductor behavior of spinel (Ni_x_Co_y_Mn_3-x-y_)O_4_ (NMC) extends its applicability to monitor the temperature in a wide range of industrial applications requiring highly accurate temperature detection: electrical resistance of the NMC is decreased with increasing temperature [1]. Especially, temperature measurements combined with fast and precise computation and the rapid development of digital circuits further enable temperature control and prediction of real-time temperature monitoring, thereby facilitating the development of a battery management system for electric vehicles, which can efficiently control battery operation and prevent safety concerns related to overheating of battery cells [2,3]. The performance of an NMC as a temperature sensor is commonly evaluated based on room temperature resistivity (*ρ*_25_) and B value. Notably, the B value represents the sensitivity to temperature change, characterized by activation energy and Boltzmann constant [1]. It is believed that polaron hopping of charge carriers among transition elements at octahedral B sites of the spinel NMC can determine *ρ*_25_ and B value for accurate temperature measurements [4,5,6,7]. For this reason, various sample preparation methods have been developed to tune the electrical properties of the spinel NMC by controlling the chemical composition and atomic structure [8,9].

The solid-state reaction (SSR) is a cost-effective and straightforward method that offers several advantages in materials synthesis and processing, including its simplicity of composition control [10,11,12]. Especially, SSR facilitates the pelletization of crystalline oxide, including NMC, thereby enabling its easy integration for industrial applications: for example, solid NMC pellets are tailored to small sensor components, which are integrated onto the printed circuit boards (PCBs) for generating a temperature sensor through a well-established packaging process [13]. SSR generally comprises two steps, calcination for the crystallization of oxides, followed by sintering for pelletization, which requires a high firing temperature above 1200 °C and a long processing time [14]. However, the manufacturing process using a two-step heat treatment requires large power consumption, resulting in increased process costs. Therefore, an efficient heat treatment route for the SSR of the NMC is needed, while maintaining the excellent performance of the NMC as a temperature sensor. 

In this work, an efficient one-step SSR route for crystalline NMC pellet, comprising calcination and sintering at a single heating profile, is proposed. The electrical properties of the NMC with temperature changes are discussed and compared to those of the NMC prepared using the one-step SSR route. Additionally, activation energies of the NMC via the two different processing routes are evaluated for their application to the temperature sensor. 

## 2. Experimental Procedure

Mn_3_O_4_ (99.9%, Junsei Chem. Japan) Co_3_O_4_ (99.9%, Junsei Chem. Japan), and NiO (99.9%, Kojundo Chem. Japan) powders as raw materials were weighed with stoichiometric amounts needed to obtain the Ni_0.3_Mn_1.5_Co_1.2_O_4_ (NMC). The oxide powders were mixed with 5% polyvinyl alcohol (PVA) in deionized water (D.I. water) via ball-milling and then dried using a spray drying process. The prepared powders were further dried at 60 °C for 12 h in the air. For applying the one-step heat treatment, the oxide powders were uniaxially pressed into pellets of 1.2 cm^2^ × 0.5 cm under a pressure of 8.3 ton/cm^2^. After drying the green body in the air for 24 h, pellets were fired in the air with a heating profile, as described in Appendix A. For applying the two-step heat treatment, the dried powders were calcined at 900 °C, 1000 °C, and 1100 °C in air, then crushed using mortar and pestle. The calcined powders were mixed with 5% PVA in D.I. water and then dried in air for 1 h. The powders were uniaxially pressed into pellets of 1.2 cm^2^ × 0.5 cm under a pressure of 8.3 ton/cm^2^, followed by further drying for 24 h. The pellets were sintered in the air, as described in Appendix A. The sintered pellet prepared via the one-step SSR route is presented in Appendix A, and Ag paste was coated on the pellet for resistance measurements with temperature. Relative densities of the samples were measured using Archimedes’ method.

Crystallographic information and microstructure of the pellets prepared via the two different processing routes were observed using the X-Ray diffractometer (XRD, Empyrean, Malvern Panalytical, England) with Cu Kα radiation (λ = 1.5404 Å) and scanning electron microscopy (SEM, JSM-7610F PLUS, JEOL, Japan), respectively. Additionally, the element distribution of the samples was analyzed using energy dispersive X-ray spectroscopy (EDS-7557, Oxford Instruments, UK). The oxidation states of the elements in the samples were measured using an X-ray Photoelectron spectrometer (XPS, NEXSA, ThermoFisher Scientific, USA). Thermal analysis was also performed using thermogravimetric-differential thermal analysis (TG-DTA, STA449, NETZSCH, Germany). To evaluate the semiconductive behavior for potential application as a temperature sensor, electrical resistance was measured in the temperature range between −15 °C and 85 °C using an LCR meter (IM3570, Hioki, Japan) after applying the silver paste on the surface of the pellets.

## 3. Results and Discussion

The effect of calcination temperature on the crystal structure of the NMC is depicted in Figure 1. As the calcination temperature increases up to 1100 °C, the main phase shifts to a mixed cubic and tetragonal spinel structure [1,15].

Previous reports indicate that the crystallographic phases of (Ni_x_Mn_y_Co_3-x-y_)O_4_ are determined by the ratios of Ni, Co, and Mn in the spinel structure. Hence, the Ni_0.3_Mn_1.5_Co_1.2_O_4_ in this study is identified as a mixture of cubic and tetragonal spinel structures [16,17]. Sintering of the NMC samples calcined at different temperatures reveals similar crystallographic phases, irrespective of the calcination temperature, as shown in Figure 2a. This indicates a mixed spinel NMC with both cubic and tetragonal structures. These observations suggest that densification of the powder compact occurs with a chemical reaction leading to the desired spinel NMC formation at elevated temperatures, up to the maximum sintering temperature. By comparing the XRD patterns of the NMC samples prepared via the two-step SSR route (including calcination and sintering) to those of the NMC prepared via the one-step SSR route (Figure 2b), a similar mixture of cubic and tetragonal spinel NMC is observed.

Microstructure evolution of the NMC powders, calcined at elevated temperatures up to 1100 °C, is displayed in Figure 3 and Figure 4, which indicates that microstructure analysis and corresponding EDS maps of the NMC samples prepared via both one-step and two-step SSR routes demonstrate a dense microstructure with uniform distribution of Co, Mn, and Ni throughout the microstructure.

Additionally, the volume density of the NMC prepared via the two-step SSR is increased up to 4.7799 g/cm^3^, which is similar value to the NMC prepared by the one-step SSR (4.6013 g/cm^3^). This suggests that the one-step SSR route effectively provides sufficient energy not only for the formation of the desired crystalline NMC with a spinel structure but also for the densification of the NMC compact. The microstructure of the NMC prepared via the two-step SSR route with different calcination temperatures is presented in Appendix A.

To understand the chemical reaction and sintering behavior during the one-step SSR route, thermo-gravimetric analysis was performed, and the result is shown in Figure 5a. Initial weight loss was observed in the NMC when the temperature was increased to 390 °C due to the evaporation of surface-absorbed water [18]. Further heating to 550 °C increases the weight in the NMC, accompanied by an exothermic reaction related to the oxidation of Mn^3+^ cations on octahedral sites [19]. Additionally, weight loss with endothermic reaction at 945 °C provides strong evidence that the formation of spinel NMC structure occurs. Further heating to 1250 °C leads to gradual weight loss, which implies that sintering takes place for the densification of the NMC pellet. For comparison, TG-DTA data of the calcined NMC at elevated temperature is also presented in Figure 5b.

Negligible weight loss was observed when the temperature was increased to 1100 °C, which indicates that chemical reaction for the formation of the spinel NMC does not occur. However, gradual weight loss from 1200 °C was observed, similar to the one observed in the NMC prepared via the one-step SSR route. Based on the TG-DTA results, it is evident that the chemical reaction for the formation of spinel NMC and following sintering occurs during the one-step SSR route, which implies that power consumption and the processing time are significantly reduced by the one-step SSR route for the fabrication of the crystalline spinel NMC pellet.

Applying different thermal energy during heat treatment can influence the arrangement of constituent ions at the A and B sites of the spinel NMC, and thus the varying valence states of the ions [20]. To investigate the composition and chemical states of the elements, XPS was performed, and the results are shown in Figure 6.

As shown in Figure 6a–c, the Co 2p spectra of the NMC via a two-step SSR route show two major peaks at 780.1 eV and 796 eV, corresponding to the Co 2p_3/2_ and Co 2p_1/2_, respectively. Main peaks can be decomposed into two characteristic curves, indicating the presence of Co^2+^ and Co^3+^. The Co 2p_3/2_ of Co^2+^ and Co^3+^ appear at 781.5 eV and 779.8 eV, respectively. Additionally, the Co 2p_3/2_ and Co 2p_1/2_ satellite peaks appear at 786.7 eV and 802.2 eV, respectively [18]. Ni 2p spectra show that Ni 2p_3/2_ and Ni 2p_1/2_ energy levels are centered at 855.7 eV and 873 eV, respectively. Additionally, the satellite peaks of Ni 2p_3/2_ and Ni 2p_1/2_ are located at 879.1 eV and 860.6 eV, respectively, indicating Ni^2+^ [19]. Mn 2p_1/2_ spectra show the binding energies of 654.8 eV, 653.3 eV, and 652.6 eV, while Mn 2p_3/2_ spectra show the binding energies of 640.9 eV, 642.7 eV, and 645.6 eV, which indicates the Mn^2+^, Mn^3+^, and Mn^4+^, respectively [21]. Although the sample prepared using the one-step SSR route shows a similar degree of oxidation of elements in the NMC consisting of Co^2+^, Co^3+^, Ni^2+^, Mn^2+^, Mn^3+^, and Mn^4+^ as presented in Figure 6d–f, the peak area ratio of Mn 2p spectra shows less Mn^2+^, and more Mn^3+^ and Mn^4+^ in the NMC, compared to the NMC prepared via the two-step SSR route, as summarized in Appendix A. This indicates that the one-step SSR route can increase the number of Mn^3+^/Mn^4+^ pairs at octahedral sites in spinel structures that lower the electrical resistivity of the NMC [22,23]. Additionally, the spinel NMCs via two different processing routes in Figure 7 show similar O 1s spectra, which present the surface lattice oxygen (O_L_) at 530 eV, chemisorbed oxygen (O_C_) at 531.4 eV, and oxygen vacancies (O_V_) at 533.6 eV in the spinel structure [24]. The peak area ratio is summarized in Appendix A, and XPS spectra of the NMC via the two-step SSR route with different calcination temperatures are also presented in Appendix A.

The electrical resistivity of the NMC prepared via different processing routes as a function of temperature is shown in Figure 8.

Both NMCs prepared via the different processing routes show the electrical properties with negative temperature coefficient (NTC), as shown in Figure 8a, confirmed by the exponential decrease in resistivity with the increase in temperature from 258 K to 358 K. After sintering, the resistivity of the samples calcined at 900 °C, 1000 °C and 1100 °C shows 397.83, 589.41 and 442.35 Ω·cm, respectively, while the sample prepared via the one-step SSR route shows a significantly lower resistivity of 12.05 Ω·cm. Significantly lower resistivity of the sample prepared using one-step SSR is likely due to the increased number of Mn^3+^/Mn^4+^ pairs and a higher ratio of tetragonal to cubic spinel phase. Although the resistivity of the NMC prepared via the one-step SSR route is different from that of the NMC prepared via the two-step SSR route, the design of the sensor module modulating the form factor can easily tune the electrical resistance required from industrials. The logarithm of the resistivity is linearly related to the inverse temperature in Figure 8b, which is due to the semiconductive behavior of the NMC following the equation below:(1)$$R=R_{0}\exp\left(\frac{E_{a}}{k\cdot T}\right)=R_{0}\exp\left(\frac{B}{T}\right)$$
where, B value is the energetic constant which is directly related to the activation energy for hopping conduction (E_a_). Additionally, ρ, k, and T are resistivity, the Boltzmann constant, and absolute temperature, respectively. B value, resistivity, and temperature coefficient of the NMC depending on different processing routes are summarized in Appendix A. The general B value range for NTC thermistors-based temperature sensor can vary depending on the specific type and application. However, a common range for the B value of the sensor is between 3000 K and 5000 K, and temperature sensor developed in this study can be adopted for industrial applications [1,3,6,19]. Based on Equation (1), we calculated activation energy for the hopping conduction of the NMC, and the results are 0.3058 eV and 0.3060 eV with calculated errors of ±0.02 for the NMC prepared via one-step and two-step SSR routes, respectively. Note that activation energy reflects the energy for hopping of the charge carrier for the NMC [25,26]. The activation energies of the NMCs prepared via the one-step and two-step SSR in this study show similar values to that of the Ni-Co-Mn-based NTC thermistor [27]. In the view of industrial application, different semiconductive properties of the NTC-based temperature sensor are required for the specific application, considering product design as well as compatibility of the other electronic components. As shown in this study, controlling the electrical conductivity along with maintaining the B value of the NMC enable the researchers to easily design the form factor of the NTC-based temperature sensor.

## 4. Conclusions

Semiconductor behavior of the spinel (Ni_x_Co_y_Mn_3-x-y_)O_4_ (NMC) depending on different processing routes including one-step solid solution reaction (SSR) and two-step SSR was investigated: one-step SSR route consists of calcination and sintering at single heating profile, while two-step SSR route includes individual calcination and sintering process. Both processing routes show a dense microstructure of the NMC. Additionally, the crystallographic phase of the NMC prepared via one-step and two-step SSR routes shows a mixed spinel structure with cubic and tetragonal phases. Additionally, Ni, Co, and Mn ions of the NMC show similar valence states: Ni^2+^, Co^2+/3+^, and Mn^2+/3+/4+^ are observed in the NMC prepared via the two different processing routes. However, more Mn^3+^/Mn^4+^ pairs were observed in the spinel NMC prepared using the one-step SSR route, which can lower the electrical resistivity. The NMC prepared via the two different processing routes shows an exponential decrease in resistivity with an increase in temperature, and the logarithm of resistivity is linearly related to the inverse temperature, which indicates that NMC is applicable in the temperature sensor. Electrical resistivity at 25 °C of the NMC prepared via one-step and two-step SSR route is 12.05 Ω and 442.35 Ω, respectively. The difference in electrical resistivity can be attributed to the ratio of tetragonal to cubic spinel structure and the number of Mn^3+^/Mn^4+^ pairs induced by different processing routes. Additionally, B values of the NMC prepared using the one-step and two-step SSR routes are 3549 K and 3551 K, respectively. Based on the results, we suggest that the one-step solid solution reaction (SSR) route is an efficient processing method to prepare spinel NMC pellet as NTC-based temperature sensor. Simplified processing steps for the fabrication of electroceramics can significantly reduce power consumption and processing time, thereby not only reducing carbon emissions but also saving manufacturing costs. Additionally, it is expected that the efficient processing methodology suggested in this study can be a starting point to develop more efficient and green processing technology for the ceramic industry.

## Figures and Tables

**Figure 1 sensors-23-05380-f001:**
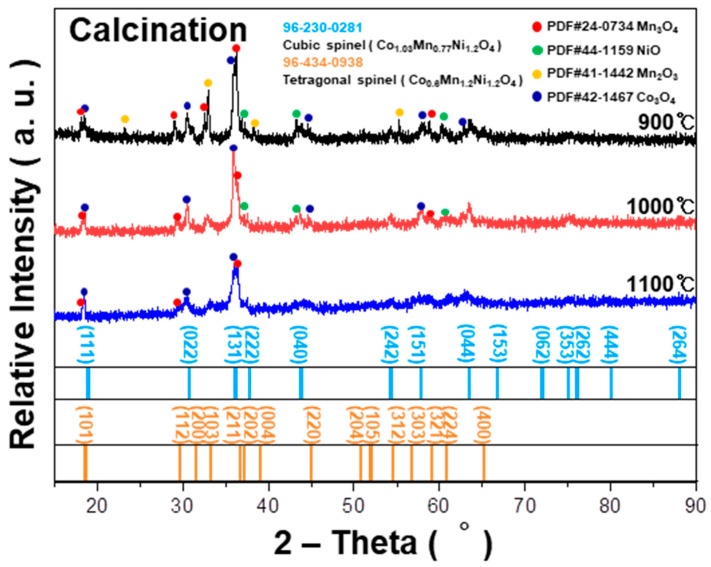
XRD patterns of NMC calcined at 900 °C, 1000 °C, and 1100 °C.

**Figure 2 sensors-23-05380-f002:**
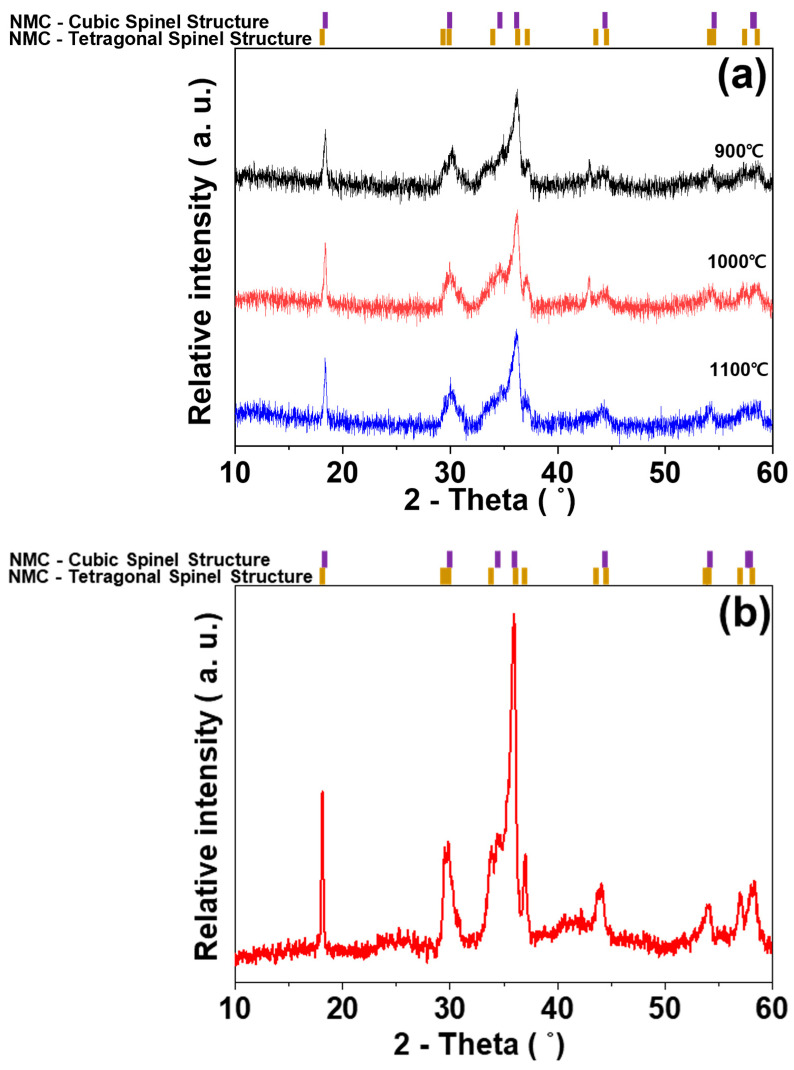
XRD patterns of NMC (**a**) via two-step SSR route at different calcination temperatures and (**b**) one-step SSR route.

**Figure 3 sensors-23-05380-f003:**
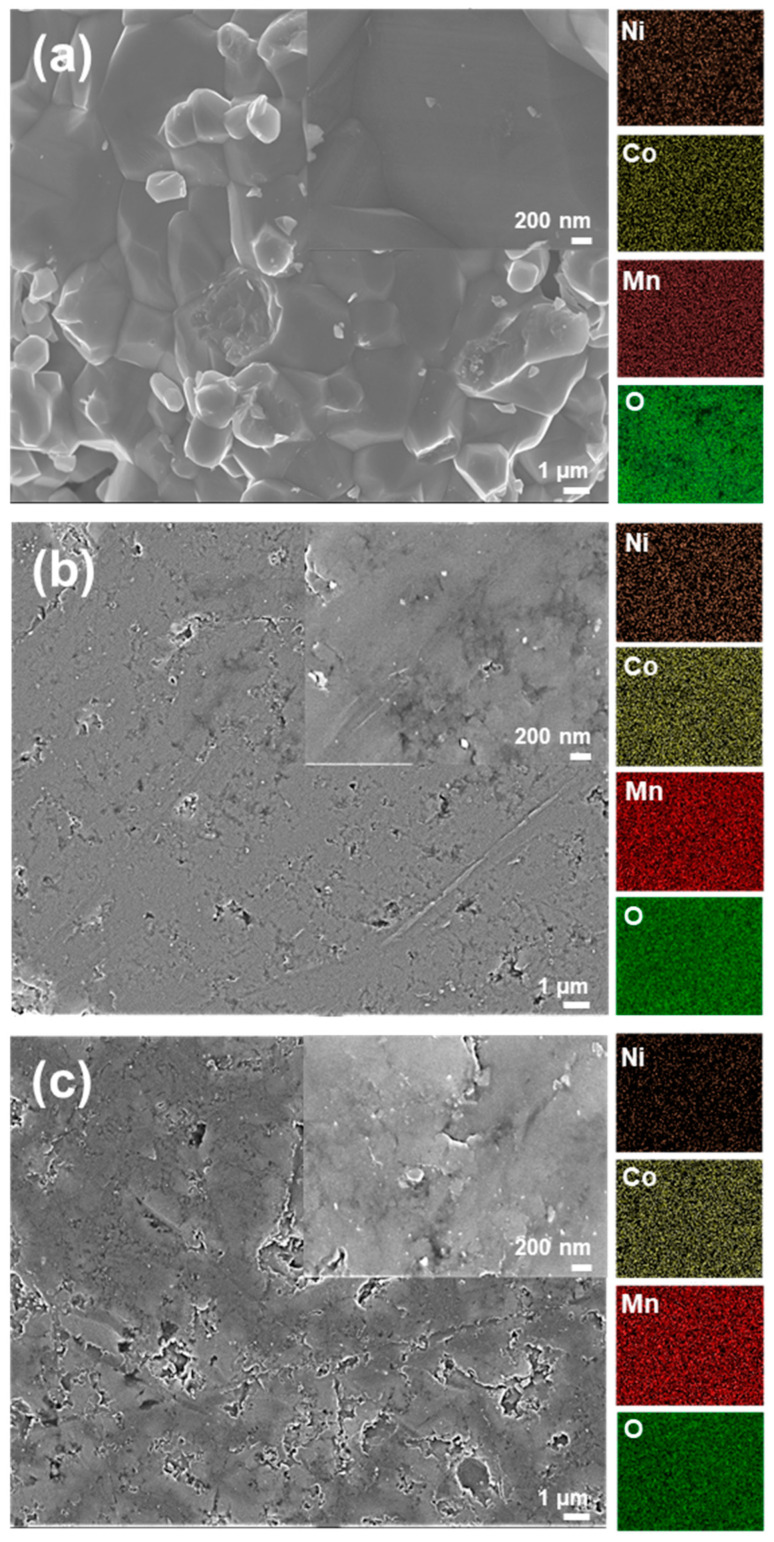
SEM images of the NMC (**a**) after calcination at 1100 °C (two-step SSR route), (**b**) after calcination at 1100 °C followed by sintering (two-step SSR route), and (**c**) after sintering (one-step SSR route).

**Figure 4 sensors-23-05380-f004:**
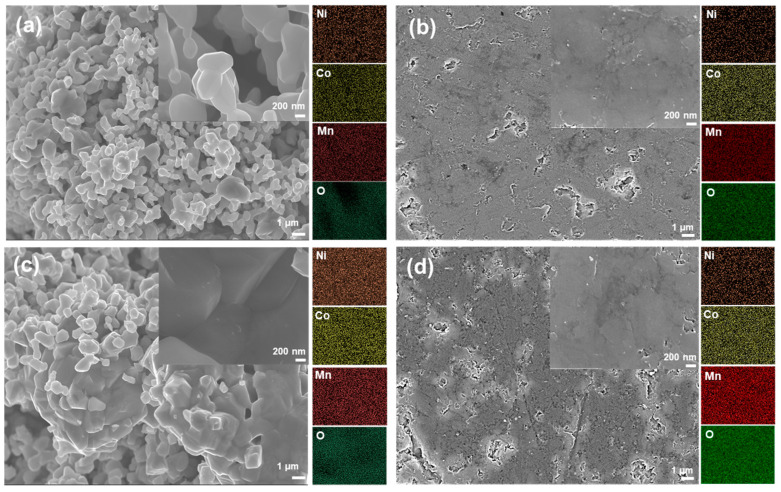
SEM images of the NMC (**a**) after calcination at 900 °C (two-step SSR route), (**b**) after calcination at 900 °C followed by sintering (two-step SSR route), (**c**) after calcination at 1000 °C (two-step SSR route), and (**d**) after calcination at 1000 °C followed by sintering (two-step SSR route).

**Figure 5 sensors-23-05380-f005:**
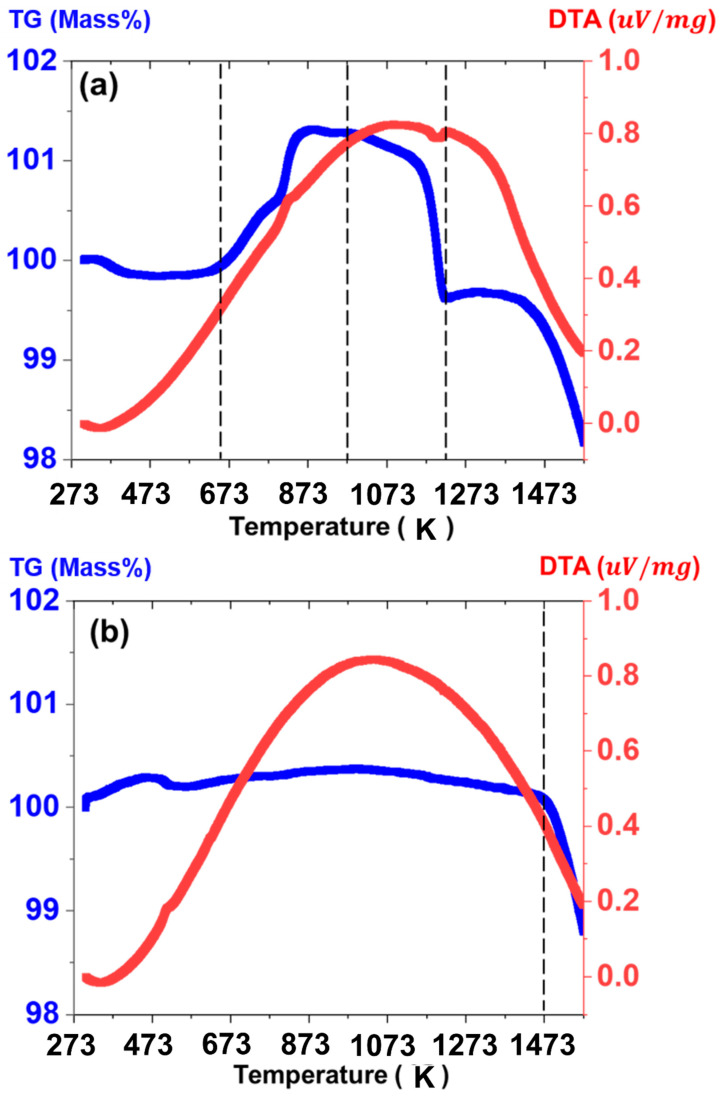
TG-DTA of the NMC via (**a**) one-step SSR route and (**b**) two-step SSR route.

**Figure 6 sensors-23-05380-f006:**
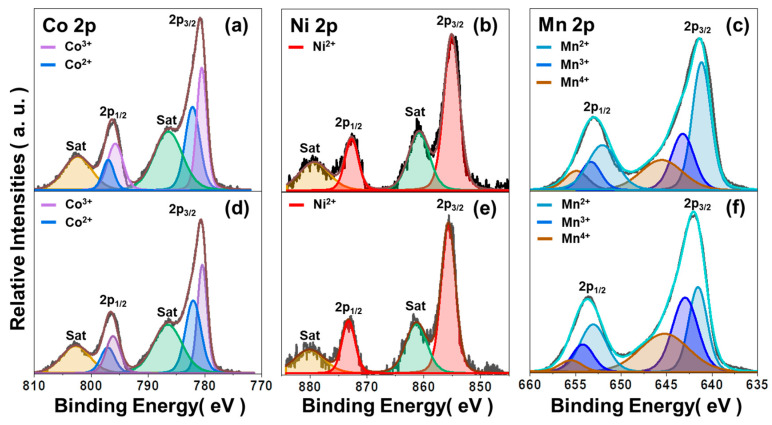
XPS spectra of (**a**) Co 2p, (**b**) Ni 2p, and (**c**) Mn 2p of the NMC via two-step SSR route (calcination temperature: 1100 °C), and (**d**) Co 2p, (**e**) Ni 2p, (**f**) Mn 2p of the NMC via one-step SSR route.

**Figure 7 sensors-23-05380-f007:**
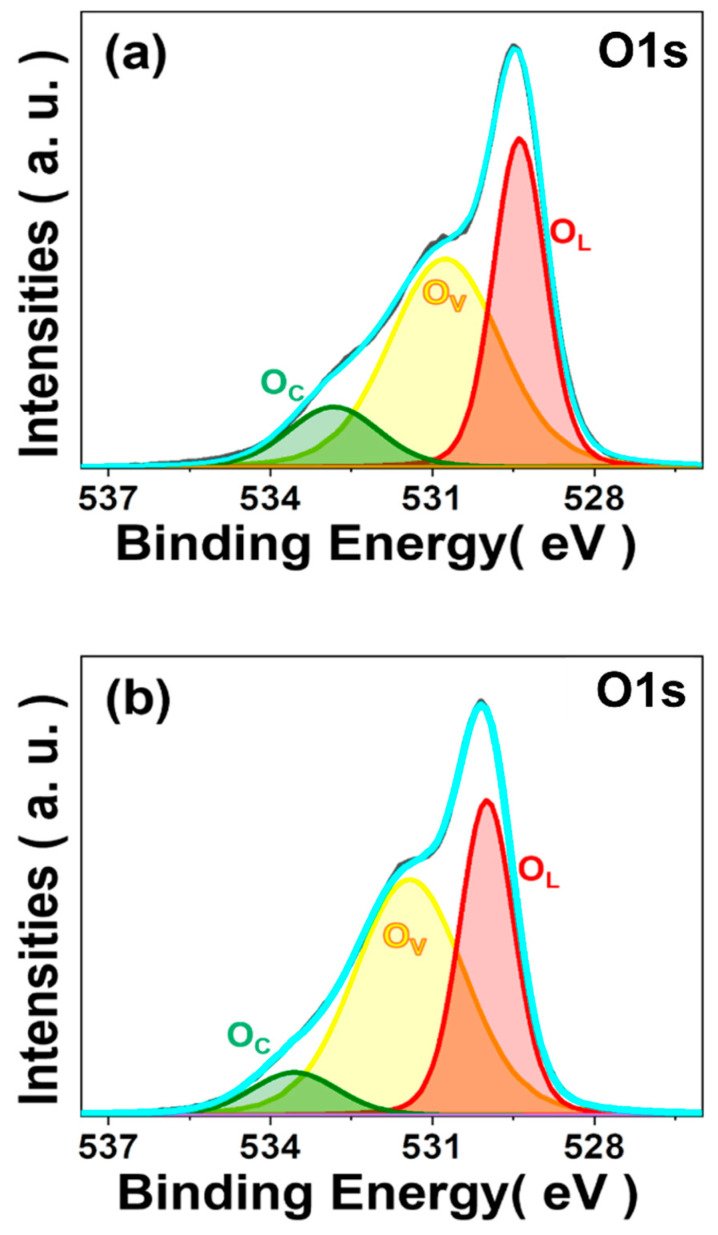
XPS spectra of O 1S of the NMC (**a**) via the two-step SSR route (calcination temperature: 1100 °C), and (**b**) via the one-step SSR route.

**Figure 8 sensors-23-05380-f008:**
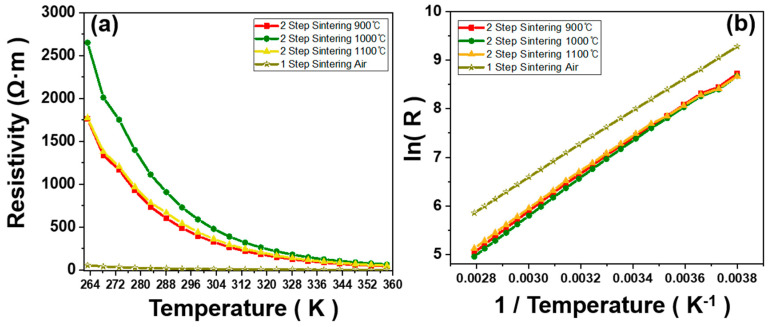
Relationship between (**a**) resistivity (ρ) and absolute temperature (T), and (**b**) ln (R) and reciprocal of absolute temperature (1/T) of the NMC.

## Data Availability

All data generated or analyzed during this study are included in this published article.

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
