# Peer review of "Effect of One Step Solid State Reaction Route on the Semiconductor Behavior of the Spinel (NI, Co, and Mn)O4 to Be Used as Temperature Sensor"

_sensors, 2023, doi:10.3390/s23125380_

Round 1

Reviewer 1 Report

In this study entitled “Effect of One-step Solid State Reaction Route on Semiconductor Behavior of the Spinel (Ni, Co, Mn)O4 as Temperature Sensor”, Ko et al. have developed a facile one-step solid solution reaction (SSR) route to produce the (Ni, Co, Mn)O4 (NMC) electroceramics. In addition, the semiconductor behavior of the one-step and two-step derived NMC electroceramics has been systematically explored. Both products have shown a decrease of resistivity exponentially with an increase in temperature, and the logarithm of resistivity is linearly related to the inverse of temperature, which indicates that NMC can be well applicable for the temperature sensors. Also, the B values of the NMCs prepared by one-step and two-step approaches are comparable. On the whole, this study is interesting, and it provides an energy-efficient route for preparing electroceramic temperature sensor. However, there are still some issues to be addressed to meet the standard of Sensors. Therefore, a major revision is recommended. The following comments should be fully considered.  

1. The temperature coefficient of the one-step derived NMC electroceramics should be extracted.   

2. Following the above comment, the temperature coefficient and the B value of the one-step derived NMC electroceramics should be compared with those of the state-of-the-art temperature sensor materials and the two-step products.  

3. In Figure 7 (Page 9), O 1S, “S” should be lowercase.  

4. One key issue of the temperature sensors for practical application is the long-term stability. So the question is that can the one-step derived NMC electroceramics maintain stable R-T characteristic over long-term storage in air (e.g., 1 month)?

5. The solid-state reaction method has a series of advantages spanning simple equipment, low cost, and wide applicability. Page 1, prior to the sentence of “Solid state reaction (SSR) facilitates the pelletization of crystalline oxide…”, a sentence summarizing the merits of solid-state reaction method along with the supporting references (e.g., Materials 2023, 16, 2449; Mater. Futures 2022, 1, 035104; Materials 2021, 14, 6976) should be provided.

English is fine. 

Author Response

Dear Reviewer,

We thank you for the review of our manuscript and consideration for publication in Sensors. We have addressed the inconsistencies or inaccuracies that were noted by the reviewer (details below and highlighted in red color in the manuscript). We therefore kindly ask that you reconsider the revised manuscript for publication in Sensors.

Reviewer 2 Report

The manuscript reports the characteristics of spinel NMC temperature sensors. The topic and results are interesting. However, there are several comments have to address here:

1.      To improve the readable, the sketch of sensing device and system structure are required.

2.      The unit is centigrade in Fig. 5. However, the unit is degree kelvin in Fig. 8. Please use same unit.

3.      How about the dimension of the sensor used in the work?

4.      The reviewer noticed the temperature range of sensing is -15 to 85 oC. It is common range. What is the superior to other sensors?

Therefore, I recommend it as major revision to publish.

Author Response

Dear Reviewer,

We thank you for the review of our manuscript and consideration for publication in Sensors. We have addressed the inconsistencies or inaccuracies that were noted by treviewer (details below and highlighted in red color in the manuscript). We therefore kindly ask that you reconsider the revised manuscript for publication in Sensors.

Round 2

Reviewer 1 Report

The authors have addressed all my comments. Now it is suggested that this manuscript can be recommended for publication. 

Reviewer 2 Report

The manuscript has revised well according the reviewer's comments. Therefore, in my opinion, the article could be accepted to published.